

# Do latitudinal gradients exist in New Zealand stream invertebrate metacommunities?

Jonathan D. Tonkin[1], Russell G. Death[2], Timo Muotka[3,4], Anna Astorga[5] and David A. Lytle[1]

[1] Department of Integrative Biology, Oregon State University, Corvallis, OR, USA
[2] Institute of Agriculture and Environment, Massey University, Palmerston North, New Zealand
[3] Department of Ecology, University of Oulu, Oulu, Finland
[4] Natural Environment Centre, Finnish Environment Institute, Oulu, Finland
[5] Institute of Ecology and Biodiversity, P. Universidad Catolica de Chile & Centro de Investigación de Ecosistemas de la Patagonia, Coyhaique, Chile

Corresponding author
Jonathan D. Tonkin,
jdtonkin@gmail.com

## ABSTRACT

That biodiversity declines with latitude is well known, but whether a metacommunity process is behind this gradient has received limited attention. We tested the hypothesis that dispersal limitation is progressively replaced by mass effects with increasing latitude, along with a series of related hypotheses. We explored these hypotheses by examining metacommunity structure in stream invertebrate metacommunities spanning the length of New Zealand's two largest islands (~1,300 km), further disentangling the role of dispersal by deconstructing assemblages into strong and weak dispersers. Given the highly dynamic nature of New Zealand streams, our alternative hypothesis was that these systems are so unpredictable (at different stages of post-flood succession) that metacommunity structure is highly context dependent from region to region. We rejected our primary hypotheses, pinning this lack of fit on the strong unpredictability of New Zealand's dynamic stream ecosystems and fauna that has evolved to cope with these conditions. While local community structure turned over along this latitudinal gradient, metacommunity structure was highly context dependent and dispersal traits did not elucidate patterns. Moreover, the emergent metacommunity types exhibited no trends, nor did the important environmental variables. These results provide a cautionary tale for examining singular metacommunities. The considerable level of unexplained contingency suggests that any inferences drawn from one-off snapshot sampling may be misleading and further points to the need for more studies on temporal dynamics of metacommunity processes.

## INTRODUCTION

The latitudinal diversity gradient is among the most well-known patterns in ecology (*Hillebrand, 2004*; *Jocque et al., 2010*). While general patterns of increasing richness from

the poles to the equator are common, there are many exceptions (*Gaston & Blackburn, 2000*; *Hillebrand, 2004*; *Heino, 2011*). The potential mechanisms behind this gradient are broad, whether non-biological (e.g. mid-domain effect hypothesis; *Colwell & Lees, 2000*), ecological (e.g. species-energy hypothesis; *Currie, 1991*), or evolutionary/historical (e.g. evolutionary rate and effective evolutionary time hypotheses; *Mittelbach et al., 2007*), but incorporating variation among local communities can provide additional insight (*Qian & Ricklefs, 2007*; *Qian, Badgley & Fox, 2009*; *Leprieur et al., 2011*; *Astorga et al., 2014*). As a key mechanism behind the latitudinal diversity gradient, climate increases in harshness with increasing latitude (*Stevens, 1989*). However, many other factors influence local climate including island size and the level of isolation. Isolated oceanic islands, for instance, have lower seasonality and predictability than continental locations at similar latitudes (*Tonkin et al., 2017a*; Fig. 1). *Jocque et al. (2010)* argue that a shift in climatic stability with latitude drives a dispersal–ecological specialisation trade-off at the metacommunity level, producing gradients in dispersal ability, ecological specialisation, range size, speciation, and species richness. In particular, increased temporal variability in environmental conditions promotes increased dispersal ability of organisms (*Dynesius & Jansson, 2000*; *Jocque et al., 2010*).

Community differences attributable to latitude are therefore likely to be driven by underlying metacommunity processes. Four metacommunity archetypes have been synthesised to summarise the relative roles of local (niche) and regional (dispersal) processes in community assembly (*Leibold et al., 2004*; *Holyoak et al., 2005*; *Leibold & Chase, 2018*): neutral, patch dynamics, species sorting, and mass effects. What remains to be tested, however, is the influence that latitude has on the roles of different metacommunity processes (*Jocque et al., 2010*). In a testable hypothesis, *Jocque et al. (2010)* predicted a stronger role of dispersal limitation in the tropics accompanied by a shift to more species sorting and mass effects with increasing latitude.

Situated at mid-latitudes, New Zealand comprises a series of islands spanning a large latitudinal gradient. With a climate reflecting its oceanic position, rainfall (Fig. 1) and river flow regimes are typically unpredictable (*Winterbourn, Rounick & Cowie, 1981*; *Winterbourn, 1995*). Although most streams tend to be perennial, the high variability in rainfall (*Heine, 1985*) produces considerable variation in flows, with frequent, but typically short-duration, spates and floods (*Duncan, 1987*). Coupled with their flashy flow regimes comes a lack of seasonality in some food resources because of a predominantly evergreen flora (*Winterbourn, Rounick & Cowie, 1981*; *Thompson & Townsend, 2000*). These factors, combined with its highly dynamic geological history, making the country particularly sensitive to sea-level fluctuations during the Quaternary, ultimately lead to a largely generalist, opportunistic, and seasonally asynchronous stream fauna adapted to coping with these harsh conditions and climatic unpredictability (*Winterbourn, Rounick & Cowie, 1981*; *Winterbourn, 1995*; *Thompson & Townsend, 2000*). Most notably, New Zealand streams feature a predominance of endemic genera, invertebrates with poorly synchronised and flexible life histories, and a predominance of non-specialist 'collector-browser' species (*Winterbourn, 1995*). Consequently, New Zealand stream
## A. New Zealand

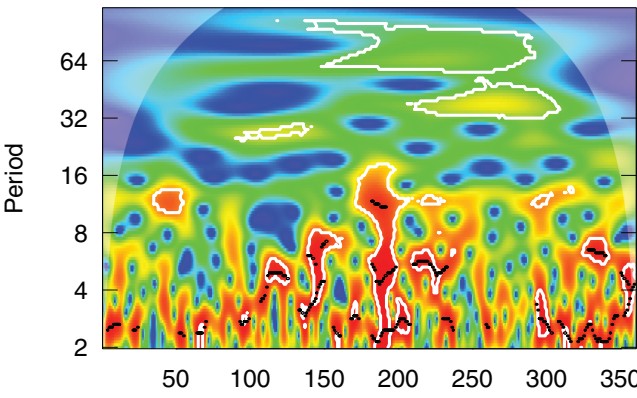

## B. Western Australia

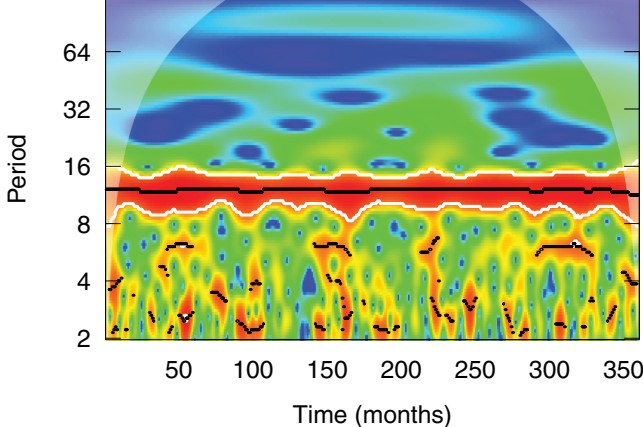

**Figure 1** **Wavelet diagram comparing 30-year monthly rainfall values between central North Island New Zealand (A) and Mediterranean-climate Western Australia (B).** The *x*-axis represent the full time series of 30 years. The *y*-axis represents the range of frequencies (period) examined within the time series. Thus the plot shows power as a function of frequency over time. Wavelet power increases from blue (low power) to red (high power). Higher power represents greater strength of the periodicity. The figure illustrates a clear, repeatable annual rainfall cycle in Western Australia (i.e. strong and consistent power at the 12-month period over the full 30-year cycle) representative of its Mediterranean climate. This contrasts to the highly unpredictable rainfall cycles in New Zealand. Wavelet analysis was performed using the R package 'WaveletComp' (*Roesch & Schmidbauer, 2014*).

communities provide an interesting test case for investigating latitudinal controls on community structure.

To test a series of hypotheses related to metacommunity structuring across a broad latitudinal gradient, we explored gradients of stream invertebrate metacommunity structure (spatial structuring and environmental filtering) spanning the length of New Zealand's two largest islands (~1,300 km). As a secondary exploration, we examined the best-fit idealised 'metacommunity types' assigned through the Elements of Metacommunity Structure framework (EMS; *Leibold & Mikkelson, 2002*). To further

disentangle the role of dispersal, we deconstructed assemblages into strong and weak dispersers. Doing so can be fruitful for exploring processes behind latitudinal diversity gradients (*Kneitel, 2016*). Taking a multi-faceted approach across latitudinal gradients allows for identifying complementary patterns in factors shaping metacommunities, compared to local community structure, advancing our understanding of how communities assemble in such dynamic landscapes.

We tested the following primary hypotheses based on the predictions of *Jocque et al. (2010)*: Metacommunities are primarily structured by environmental variables (in line with the species sorting archetype; $H_{1a}$) and spatial variables increase in importance from north to south (representing increasing dispersal and in line with the mass effects archetype; $H_{1b}$). The alternative to this hypothesis ($H_1A$) is that, given the highly dynamic nature of New Zealand streams (*Winterbourn, Rounick & Cowie, 1981*), they are so unpredictable (at different stages of post-flood succession) that metacommunity structuring is context dependent from region to region. Because environmental heterogeneity and the spatial extent of metacommunities are important regulators of the relative strength of species sorting compared to dispersal limitation and surplus (both of which should increase the spatial signature in the metacommunity) (*Heino et al., 2015b*), we also explored the influence of these factors on observed patterns. Using the deconstructed dispersal groups, we tested the secondary hypothesis, based on the predictions of *Jocque et al. (2010)* ($H_2$), that strong dispersers increase from north to south. The EMS analysis was used as an additional exploratory analysis, thus we did not form any specific hypotheses.

## METHODS

### Study sites

We used data previously collected (*Astorga et al., 2014*) from 120 streams in eight regions (15 sites in each region), spanning a latitudinal gradient of 12° (Fig. 2). Values of regional γ and β diversity, and mean α diversity are reported in *Astorga et al. (2014)*. These eight datasets span across the five biogeographic regions of the New Zealand mainland (*Di Virgilio et al., 2014*): two in northern North Island, two in southern North Island, one in central New Zealand, two in mid-South Island, and one in southern South Island. Site selection followed a series of criteria, outlined in the following sentences, to minimise differences between regions. Streams were sampled primarily in protected areas (National or State Forest Parks) and were restricted to those with maximum of 14% exotic forestry and 30% pasture in the upstream catchment. All sites had a minimum intact riparian buffer of 50 m (Freshwater Ecosystems of New Zealand (FENZ)) (*Leathwick et al., 2010*) and were selected in proportion to FENZ classes in regions. Sites were restricted to <7 m wide headwater streams (order 1–3), with similar aspect and with a rocky substrate, and sampling was confined to the riffle zone. Almost all of New Zealand's landmass belongs to the temperate oceanic (Cfb) climate zone (*Peel, Finlayson & McMahon, 2007*). Although the large majority of our sampling sites were geographically situated in this zone, some of the South Island sites (e.g. in Fiordland and Arthur's Pass) likely fell on the border of temperate oceanic, subpolar oceanic (Cfc) and tundra (ET) zones.

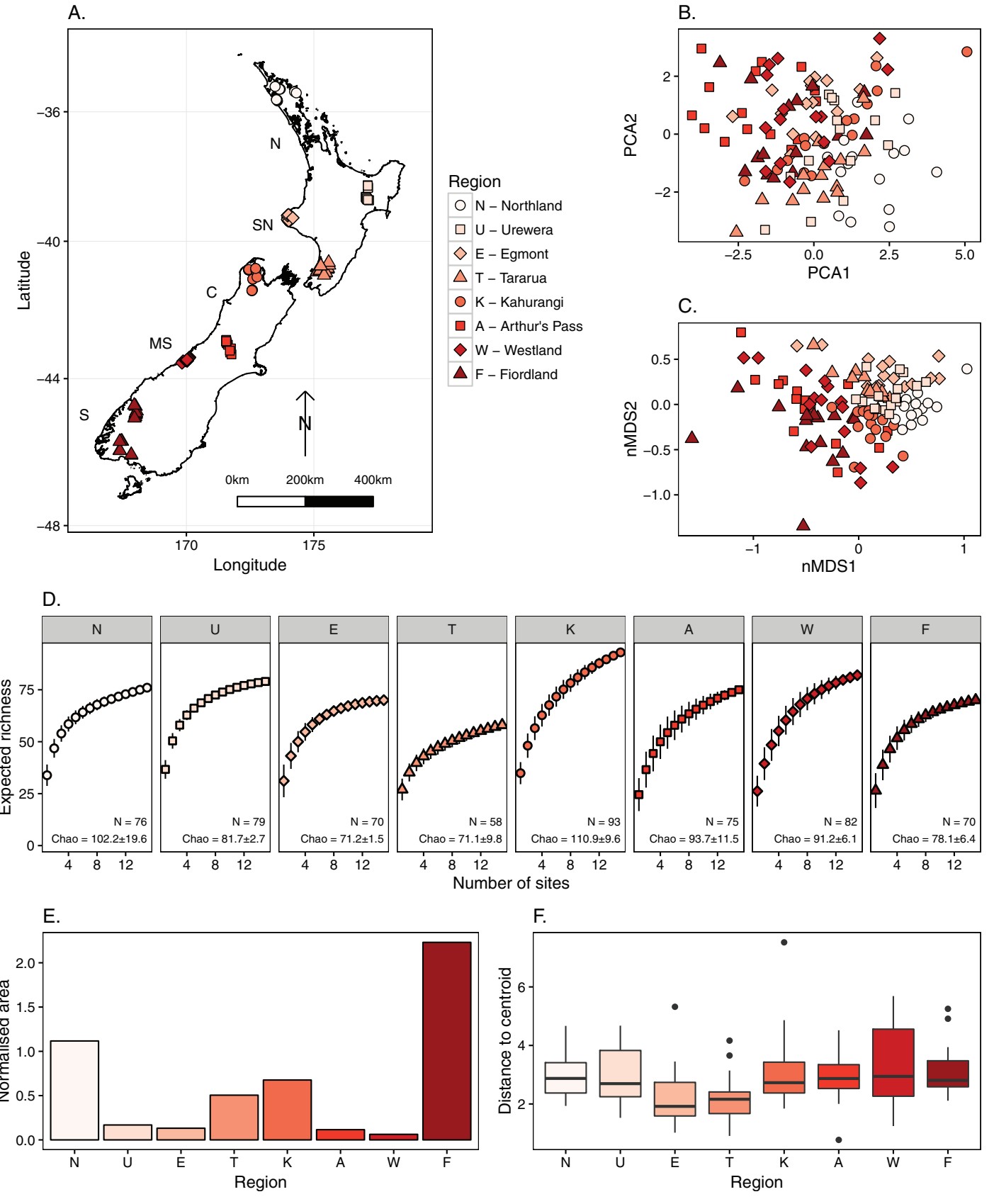

**Figure 2 Overview of sites and regional invertebrate assemblages across New Zealand.** All plots are colour-coded and shaped in the same manner, from north to south. (A) Distribution of 120 sites across eight regions of New Zealand. The five biogeographic regions are displayed as letters alongside the plot (N, Northern North Island; SN, Southern North Island; C, Central New Zealand; MS, mid-South Island; S, Southern South Island). (B) First two components of principal component analysis on environmental variables used in the study. Proportion of variation explained: PCA1 = 0.21; PCA2 = 0.17. (C) Non-metric multidimensional scaling ordination of invertebrate communities from all 120 sites. 2D stress = 0.21. (D) Species accumulation curves for all species for the eight regions. Regions are ordered from north (left) to south (right). Displayed text shows sampled regional richness (N) and Chao's estimate of total regional richness with standard error. (E) Spatial extent of each metacommunity (normalised area). (F) Environmental heterogeneity of each metacommunity, measured through homogeneity of dispersions.

Nevertheless, all sites were situated in areas without shortage of rainfall, which has been described as 'plentiful' but temporally variable in New Zealand (*Heine, 1985*), although there are more arid regions in eastern zones such as Hawke's Bay and central Otago. Therefore, all sites had permanent flow and the large majority of streams were runoff fed.

## Benthic macroinvertebrate sampling

Benthic macroinvertebrate sampling was performed between February and April 2006 (Austral summer/autumn) using 2 min kick-net (0.3 mm mesh) samples. Kicks were performed with the goal of covering most of the microhabitats present in a ca. 100 m$^2$ riffle section. This approach captures ca. 75% of the benthic invertebrate species at a site, covering 1.3 m$^2$ of the benthos (*Mykra, Ruokonen & Muotka, 2006*). Samples were stored in 70% ethanol and later sorted and identified to the lowest possible taxonomic level (usually genus or species, but certain difficult-to-identify species, such as chironomid midges were left at higher taxonomic levels), following *Winterbourn, Gregson & Dolphin (2000)*.

To help understand the role of dispersal (inherent in all of our hypotheses), we focused our analysis on three data matrices: all species combined, species with high dispersal ability, and species with low dispersal ability. These dispersal ability groups were assigned based on pre-defined trait categories established for New Zealand aquatic invertebrates (*Doledec et al., 2006*; *Doledec, Phillips & Townsend, 2011*). Such a deconstruction approach is commonly applied in riverine metacommunity studies, and can help to disentangle the effects of dispersal (*Tonkin et al., 2018a*). However, these dispersal traits do not necessarily reflect actual dispersal rates (*Lowe & McPeek, 2014*; *Lancaster & Downes, 2017*). The analyses that follow used a combination of log- or Hellinger-transformed abundance data or presence–absence data on a case-by-case basis, which we specify below.

## Environmental variables

We included several previously identified important local habitat variables for stream invertebrate communities (*Tonkin, 2014*; *Astorga et al., 2014*; *Tolonen et al., 2017*), as well as stream order and elevation in our analyses (Table 1). Local habitat variables were as follows: water temperature, electrical conductivity, pH, wetted width, reach slope, water depth, overhead canopy cover, periphyton biomass (chlorophyll *a*), bryophyte percent cover, Pfankuch index (bottom component), and substrate size index (SI).

**Table 1 Environmental variables used in the analysis.**

| Variable | Units | Explanation |
|---|---|---|
| Temp | °C | Water temperature |
| Cond | $\mu S\ cm^{-1}$ | Conductivity |
| pH | – | pH |
| Width | cm | Wetted width |
| Elev | m a.s.l. | Elevation |
| Slope | $cm\ m^{-1}$ | Slope of the stream reach |
| Depth | cm | Depth |
| OHCov | % | Percent overhead canopy cover |
| Chla | $\mu g\ cm^{-2}$ | Chlorophyll *a* (periphyton biomass) |
| Bryophytes | % | Percent moss cover |
| Pfankuch_bottom | – | Stream bed stability |
| SI | – | Substrate size index |
| Order | – | Stream order |

Depth was measured at 40 random locations in transects across the channel. Canopy cover was measured at 20 evenly spaced cross-channel transects with a densiometer. Channel slope was measured with an Abney level over 10–20 m. Percentage of bryophytes was visually estimated for each reach. Substrate composition was measured by taking 100 randomly selected particles at 1 m intervals along a path 45° to the stream bank in a zig–zag manner. Particles were assigned to each of 13 size classes: bedrock, >300, 300–128, 128–90.5, 90.5–64, 64–45.3, 45.3–32, 32–22.6, 22.6–16, 16–11.3, 11.3–8, 8–5, and <5 mm. These were then converted to a single substrate size index (SI) by summing the mid-point values of each size class weighted by the number of stones in each class (bedrock was assigned a nominal size of 400 mm).

Stream bed stability was assessed with the bottom component of the Pfankuch Stability Index (*Pfankuch, 1975*). The Pfankuch Index is a visual assessment method designed to give an index of channel stability. The index can be broken down into three individual components: upper banks, lower banks, and stream bed (bottom). We used the bottom component as it is the most relevant to stream invertebrates (*Schwendel et al., 2012*). The bottom component consists of six wetted channel attributes (substrate brightness, angularity, consolidation, percentage of stable materials, scouring, and amount of clinging aquatic vegetation), which can be assigned to predetermined categories with weighted scores. The sum of these scores represents the stability of the substrate, where high values represent low stability.

As an assessment of periphyton biomass (measured as chlorophyll *a*: $\mu g\ cm^{2}$) at each site, five stones were randomly selected from the sample riffle and frozen for later analysis. Pigments were extracted in the laboratory by soaking the stones in 90% acetone for 24 h at 5 °C in the dark. Absorbances were read using a Cary 50™ Conc UV-Visible spectrophotometer, and chlorophyll *a* was calculated using the method of *Steinman & Lamberti (1996)*. Stone surface area was corrected using the method of

*Graham, McCaughan & McKee (1988)*, assuming only the top half of the stone was available for periphyton growth.

## Statistical analyses

### Summarising patterns across regions

To visualise patterns in the environmental conditions of sites, we used principal components analysis (PCA), performed with the *princomp* function, on the full suite of normalised environmental variables. Similarly, to examine patterns in macroinvertebrate communities across all 120 sites, we performed ordination with non-metric multidimensional scaling (nMDS), on $\log(x) + 1$ abundance data. We ran this using the *metaMDS* function, based on Bray–Curtis distances, in the *vegan* package (*Oksanen et al., 2013*). To test whether communities differed across the eight regions, we used PERMANOVA, based on the *adonis* function and 999 permutations in *vegan*. To compare the properties of diversity in each of our eight regions, and gain insight into how well sampled each region was, we calculated species accumulation curves using the *specaccum* function in *vegan* (*exact* method; *Ugland, Gray & Ellingsen, 2003*). To accompany these curves, we estimated total regional species richness using Chao's estimate (*Chao, 1987*), but it is important to note that this estimate is biased for open regions like those examined here.

Given the importance of spatial extent and environmental heterogeneity on metacommunity structuring, we calculated these for each metacommunity. For spatial extent, we calculated the convex hull of points making up each metacommunity using the *chull* function, followed by calculating the area of the polygon using the *Polygon* function. Therefore, spatial extent represents the total area that each metacommunity occupies on the landscape. For environmental heterogeneity, we calculated the homogeneity of group dispersions using the *betadisper* function in *vegan*, following the methods of *Anderson (2006)*.

### Metacommunity structuring and role of dispersal

$H_1$ was tested using a variation partitioning approach (*Borcard, Legendre & Drapeau, 1992*; *Peres-Neto et al., 2006*), where we disentangled the relative influence of spatial and environmental variables on metacommunity structure of the eight metacommunities ($n = 15$) using Hellinger-transformed macroinvertebrate community data. A stronger role of environmental variables in structuring metacommunities reflects a situation where species sorting is strong, whereas stronger spatial structuring (i.e. spatial variables explain community structure) could reflect either end of the dispersal spectrum from limitation to surplus. To partition variation, we used partial redundancy analysis (pRDA), a constrained ordination technique, to partition the variation into the pure components of space, environment and their shared contribution to the explanation of community structure. Variation partitioning attempts to isolate the pure effects of environmental gradients from spatial structure (i.e. environmental filtering) and the pure effects of spatial structure from environmental gradients (i.e. dispersal effects). Note, however, that if environmental and spatial variation overlap considerably, the spatial component from

variation partitioning analyses should be interpreted with caution (*Gilbert & Bennett, 2010*; *Tuomisto, Ruokolainen & Ruokolainen, 2012*). Shared remaining variation may result from interactive effects such as spatially structured environmental gradients or dispersal that is dependent on topography, for instance, but unmeasured environmental variables may also be interpreted as pure spatial effects. The environmental component in our analysis represents the set of pre-selected local habitat variables, and we represented the spatial structuring through Principal Coordinates of Neighbour Matrices (PCNM).

We created a set of spatial eigenvectors to represent the distribution of sites in space using PCNM (*Borcard & Legendre, 2002*; *Dray, Legendre & Peres-Neto, 2006*) with the *pcnm* function in the *vegan* package. PCNM transforms spatial distances between all sites in a metacommunity based on a distance matrix into rectangular data for use in constrained ordination methods. Despite the importance of the river network in structuring riverine metacommunities (*Tonkin, Heino & Altermatt, 2018*; *Tonkin et al., 2018a*), we focused on overland distance to represent spatial structuring. This is because the large majority of taxa in our dataset have an adult flight stage and can thus disperse overland, rather than being restricted to within-network dispersal. Moreover, while there can be differences in the influence of overland and watercourse distances (*Schmera et al., 2018*), such differences are often weak when considering invertebrates (*Tonkin et al., 2018a*). To create the PCNM vectors, we used geographic coordinates to create a distance matrix using Euclidean distances. PCNM vectors represent a gradient of organisation of sites at different spatial scales, ranging from large-scale to small. That is, PCNM1 represents the broadest-scale arrangement of sites, through to the last vector representing much finer arrangement. Only eigenvectors with positive eigenvalues were used in the analysis.

Prior to variation partitioning, we first ran global RDA models individually for environment (normalised local habitat, stream order, and elevation) and space (PCNM vectors), and tested for significance. We checked for collinearity in the models and excluded variables with a variance inflation factor (VIF) of greater than 10. We removed the variable with the highest VIF first and followed each model sequentially until no variables had a VIF > 10. After this, if the global model was significant, we then performed forward selection to select the most important variables. We used the *ordiR2step* function in the *vegan* package (*Oksanen et al., 2013*) to forward-select variables, which employs the approach outlined by *Blanchet, Legendre & Borcard (2008)*. The *ordiR2step* function selects variables that maximise the adjusted $R^2$ (adj. $R^2$) at each step. The stepwise procedure stops when the adj. $R^2$ begins to decline, exceeds the scope of the full model (i.e. full model adj. $R^2$), or the *P* value, which we set to be 0.05, is exceeded. If the global model was non-significant, we regarded that dataset to have an $R^2$ of 0. Only if both spatial and environmental models were significant, was variation partitioning performed between the two groups. We partitioned the variation between forward-selected environmental variables and forward-selected spatial vectors using pRDA with the *varpart* function in *vegan*, and tested significance of the pure effects of environment and space using the *RDA* function.
To test H$_2$, whether strong dispersers increase from north to south, we calculated the ratio of strong to weak dispersers in each metacommunity in full. All analyses, including the following, were performed in R version 3.1.1 (*R Core Team, 2014*).

### Elements of metacommunity structure

In addition to our core hypothesis testing, we employed the EMS framework (*Leibold & Mikkelson, 2002*) as an exploratory examination of metacommunity types along the latitudinal gradient. EMS is an approach used to explore and characterise emergent properties in a site-by-species matrix, using three metrics: (1) coherence, or the degree to which different species respond to the same environmental gradient; (2) turnover (range turnover), or the degree to which species replace each other along the environmental gradient; and (3) boundary clumping, or the amount of (dis)similarity (i.e. clumping) in species range boundaries. EMS differs from the variation partitioning approach in that it concurrently examines multiple idealised types of metacommunities, by comparing observed patterns against null expectation.

Prior to extracting these elements, the site-by-species matrix is organised in the most coherent manner using reciprocal averaging (*Gauch, Whittaker & Wentworth, 1977*). Reciprocal averaging arranges sites so that the species with the most similar distributions and sites with similar composition are closest in the matrix (*Gauch, Whittaker & Wentworth, 1977*); essentially arranging sites along a latent environmental gradient which is likely important in structuring species distributions. The ordered site-by-species matrix is then compared with random distributions through permutation of a null matrix.

Elements of metacommunity structure takes a three-step approach to measuring coherence, turnover, and boundary clumping. Only when a matrix has significantly positive coherence, can the following steps be performed. Coherence, the first step, can be differentiated into non-significant (i.e. random: species assemble independent of each other), significantly negative (i.e. checkerboard), or significantly positive (i.e. coherent). Checkerboard patterns represent distributions where species are found in avoidance of each other more often than chance. Checkerboards were originally thought to reflect competitive exclusion (*Diamond, 1975*), but can also represent a host of other causes such as environmental heterogeneity (*Gotelli & McCabe, 2002*; *Boschilia, Oliveira & Thomaz, 2008*). At each of the steps, the observed ordinated site-by-species matrix is compared with a null distribution. The matrix is reshuffled based on a predefined algorithm and constraints and permuted a set number of times. The observed value is then compared with the null.

Coherence is calculated through the number of embedded absences in the ordinated matrix. Embedded absences are gaps in the species range (*Leibold & Mikkelson, 2002*). If there are more embedded absences than expected by chance (i.e. through the null matrix), a metacommunity is considered checkerboarded, and vice versa (i.e. fewer embedded absences than chance). If there is no difference in the observed matrix from chance (null), random assembly is expected. For comparability, both coherence and turnover are tested using the standardised *z*-test. Coherent distributions suggest communities are structured along an environmental gradient, either individualistically or

in groups. Turnover and boundary clumping are then examined on the positively coherent distributions.

The turnover step enables differentiation into the set of gradient models that best fit the data structure. Turnover is measured as the number of times a species replaces another between two sites in the ordinated matrix. Significantly negative turnover points to nestedness in distributions (further described below), whereas significantly positive can be differentiated into Clementsian, Gleasonian or evenly spaced gradients. These latter three can be distinguished based on the level of boundary clumping in species distributions, using Morista's Index (*Morista, 1971*) and an associated Chi$^2$ test comparing observed and null distributions. Values significantly greater than one point to clumped range boundaries (i.e. Clementsian gradients), less than one point to hyperdispersed range boundaries (i.e. evenly spaced gradients), and no difference from one points to random range boundaries (i.e. Gleasonian gradients). Nested subsets are also broken down based on their boundary clumping into clumped, hyperdispersed and random range boundaries.

Rather than adopt the approach of *Presley, Higgins & Willig (2010)*, where non-significant turnover is further examined into quasi-turnover and quasi-nestedness, we treated non-significant turnover as a non-structure given that it indicates no difference from the null expectation. Eight possible metacommunity types result: random, checkerboard, Gleasonian, Clementsian, evenly spaced, nested clumped, nested random, and nested evenly spaced. Detailed explanation and diagrammatic representations of these structures are available in several sources (*Leibold & Mikkelson, 2002*; *Presley, Higgins & Willig, 2010*; *Tonkin et al., 2017b*).

We constrained our null models using the fixed-proportional 'R1' method (*Gotelli, 2000*), which maintains site richness, but fills species ranges based on their marginal probabilities. The R1 null model is realistic from an ecological perspective, given that richness of a site varies along ecological gradients (*Presley et al., 2009*). Consequently, the R1 null model is recommended in the EMS analysis as it is relatively insensitive to type I and II errors (*Presley et al., 2009*). Other methods can incorporate too much or too little biology into the null model and can be thus prone to type I and II errors (*Gotelli, 2000*; *Presley et al., 2009*). Using the R1 null model, generated in the *vegan* package (*Oksanen et al., 2013*), we produced 1,000 simulated null matrices for each test. We evaluated EMS on presence–absence data, using the R package *Metacom* (*Dallas, 2014*), across the eight metacommunities individually and restricted our examination to the primary axis of the RA ordination as this represents the best arrangement of matrices. Prior to running the EMS analysis, we removed all species that were present in less than two sites, as rare species can bias the EMS results, particularly coherence and boundary clumping (*Presley et al., 2009*).

## RESULTS

The Fiordland and Northland metacommunities had the greatest spatial extents (Fig. 2E), but there was little difference in environmental heterogeneity between the regions (Fig. 2F). The gradient in environmental conditions was weak across the eight regions,

with a low percentage of variance explained (37%) by the first two principal components (Fig. 2B), and no variables contributing more than 15% to either of the first two components. Invertebrate communities differed significantly between the eight regions, with a clear latitudinal trend in assemblage structure (PERMANOVA: $F_{7,112} = 7.30$, $R^2 = 0.313$, $P = 0.001$; Fig. 2C). Regional richness tended to be highest at the north of each island and decline towards the southern zones (Fig. 2D), as demonstrated in *Astorga et al. (2014)*. The regional pool of most regions were well sampled. However, Kahurangi did not reach a clear asymptote and had the steepest species accumulation curve. Moreover, the North Island regions' curves tended to be less steep compared to those in the South Island. However, Chao's estimated values did not differ in a systematic manner, with differences between sampled and projected richness not being consistently higher in the South Island.

## Metacommunity structuring and the role of dispersal

There was no gradient with latitude in the relative importance of environmental or spatial control for all species combined and for individual dispersal groups (Fig. 3) suggesting $H_1$ can be rejected. The influence of spatial extent and its interaction with dispersal ability did not resolve this lack of pattern in the relative role of spatial or environmental components in the variation partitioning models (Figs. 2–4). Finally, contrary to $H_2$, the ratio of strong to weak dispersers decreased from north to south (Fig. 4).

When considering all species together, only three of the eight regions were significantly structured by both environmental and spatial components together, and thus could be considered for variation partitioning (Fig. 3). In the deconstructed dispersal group datasets, only one of the eight regions had combined significant environmental and spatial components. Environmental control was more commonly important than spatial in structuring both strong and weak disperser metacommunities. Northland exhibited no spatial or environmental structure for any of the datasets.

Considering all models (including those assigned 0% explained), environmental variables explained more of the variation when the whole community was considered (mean adj. $R^2 = 0.134$; 13.4% variance explained) compared to breaking into high (7.1%) and low (4.8%) dispersal ability groups. This result was particularly evident for certain regions, such as Westland, which could be explained well when considering the full community (strongest model), but not for the dispersal groups. However, strong dispersers had on average higher adj. $R^2$ values (adj. $R^2 = 0.191$; 19.1% explained) when only considering the significant models, than all combined (18.0%) or weak dispersers (9.6%). Spatial variables explained less of the variation in community structure than environmental, when non-significant models were included (adj. $R^2$—all: 0.047; high: 0.049; low: 0.054) but not when only considering significant models (adj. $R^2$—all: 0.126; high: 0.200; low: 0.143).

Forward-selected environmental variables were highly variable in the RDA models, with no particular variable consistently important across the eight metacommunities (Table 2; Table S1 in Appendix S1).

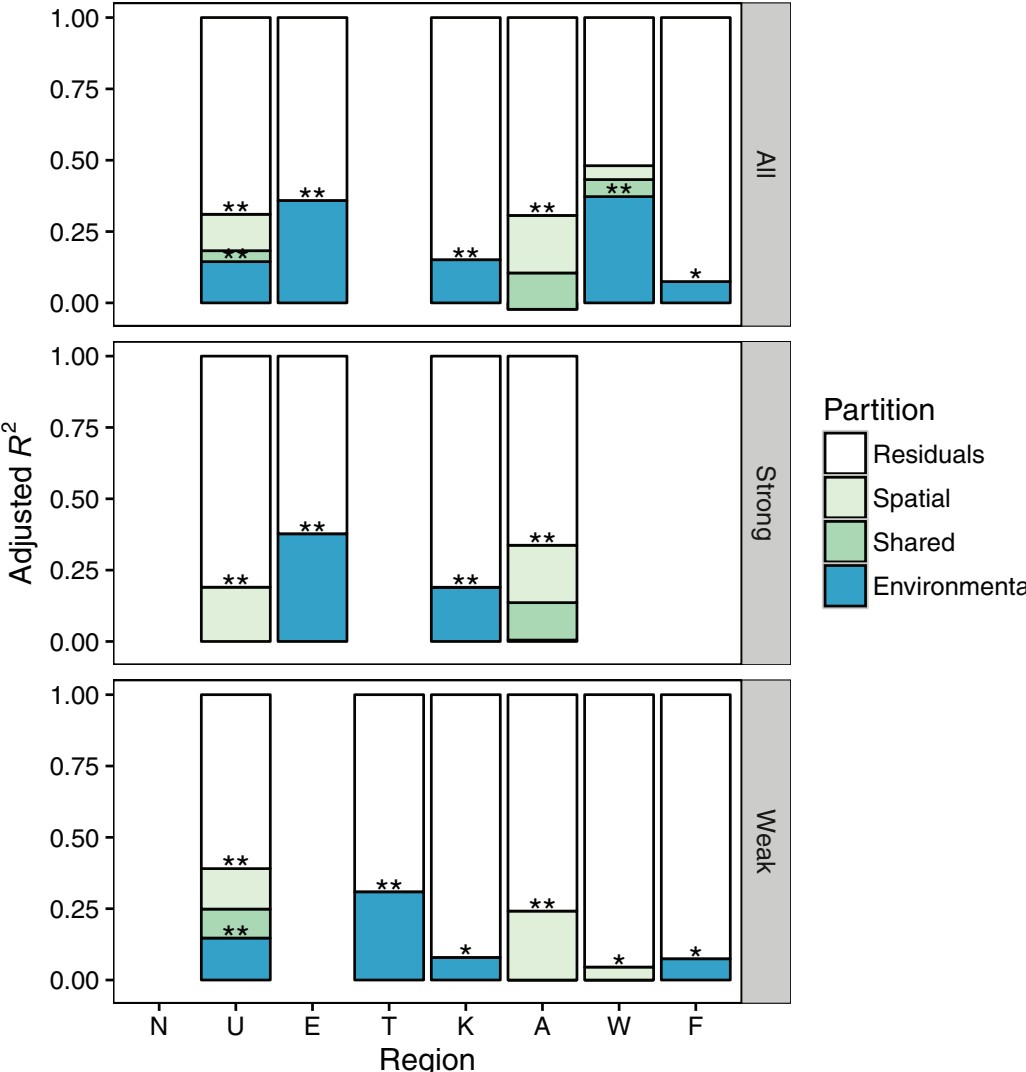

**Figure 3 Results of variation partitioning of spatial and environmental variables on macroinvertebrate communities in eight regions spanning the length of New Zealand's two largest islands.** Regions are ordered from north (left) to south (right). Variation partitioning was performed only where global RDA models were significant. Certain regions had non-significant global models for either spatial, environmental or both. Where either spatial or environmental was significant, we plot the results of the global model (and its significance). Significance of the pure effects of space or environment are shown with asterisks. All, all species; strong, strong dispersers; weak, weak dispersers.

## Metacommunity types (EMS)

There was no latitudinal trend in metacommunity type for all organisms combined and for each of the dispersal ability groups (Table 3). For the full community dataset, Gleasonian gradients were the most common pattern (five regions), indicating positive coherence and turnover, but no boundary clumping. The remaining regions' metacommunity types consisted of two regions with random structures and one with no structure (non-significant turnover). Clementsian gradients were more common for strong dispersers, with the remaining regions having either random (two regions),

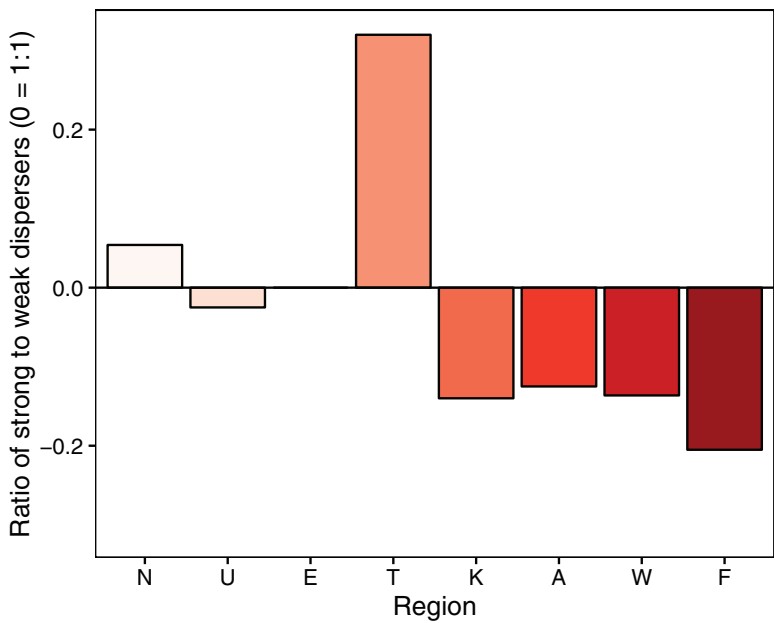

**Figure 4 Ratio of strong to weak dispersers in each metacommunity.** 0 = 1:1 ratio of strong to weak dispersers. Above the line represents a higher strong to weak disperser ratio.

**Table 2 Forward-selected environmental variables used in the variation partitioning analysis when a global RDA model was significant.**

| Subset | Region | F | P | Variables |
|--------|--------|------|-------|-----------|
| All | U | 2.57 | 0.001 | Temp, pH |
| All | E | 2.96 | 0.001 | OHCov, Elev, SI, Depth |
| All | K | 2.25 | 0.001 | Cond, OHCov |
| All | A | 2.64 | 0.026 | Temp |
| All | W | 4.55 | 0.001 | Cond, pH, Slope |
| All | F | 2.13 | 0.01 | Order |
| Strong | E | 3.83 | 0.001 | OHCov, Elev, SI |
| Strong | K | 2.64 | 0.005 | Cond, Chla |
| Strong | A | 3.20 | 0.037 | Temp |
| Weak | U | 3.32 | 0.001 | Temp, pH |
| Weak | T | 2.57 | 0.001 | OHCov, Pfankuch_bottom, Chla, Depth |
| Weak | K | 2.20 | 0.024 | Cond |
| Weak | F | 2.13 | 0.018 | Order |

**Note:**
Only if a global model was significant, was forward selection performed. Forward-selected variables are given in the 'Variables' column. Subset, subset of species (all species, and strong and weak dispersers). Full results of both global and forward-selected models, including spatial variables can be found in Table S1.

Gleasonian or no structure (non-significant turnover; Table 3). Weak dispersers were much more variable between the regions, often with weaker coherence. In fact, four regions exhibited random distributions represented by non-significant coherence. The remaining regions had either Gleasonian (two regions), Clementsian or no structure.

**Table 3 Results of Elements of Metacommunity Structure analysis examining the best-fit idealised metacommunity structure for each metacommunity, including the strong and weak disperser groups.**

| Subset | Region | df | Coherence | | | | | Turnover | | | | | Boundary clumping | | Structure |
|--------|--------|-----|-----|------|------|------|--------|--------|--------|--------|-------|--------|------|--------|-----------|
| | | | Abs | Mean | SD | z | P | Re | Mean | SD | z | P | MI | P | |
| All | N | 58 | 305 | 321.1 | 15 | 1.07 | 0.2835 | 2,148 | 1,649.8 | 580.7 | −0.86 | 0.3909 | 1.17 | 0.3468 | Random |
| All | U | 68 | 277 | 386.6 | 17.7 | 6.18 | <0.0001 | 9,768 | 2,659.4 | 823.4 | −8.63 | <0.0001 | 0.85 | 0.3928 | Gleasonian |
| All | E | 62 | 248 | 367.8 | 19.1 | 6.26 | <0.0001 | 10,931 | 2,978.5 | 980.3 | −8.11 | <0.0001 | 0.68 | 0.2683 | Gleasonian |
| All | T | 42 | 168 | 197.6 | 12.8 | 2.32 | 0.0204 | 1,334 | 1,095.8 | 388.2 | −0.61 | 0.5394 | 2.12 | 0.0044 | – |
| All | K | 66 | 325 | 384.8 | 19.5 | 3.06 | 0.0022 | 6,293 | 3,145.3 | 950.0 | −3.31 | 0.0009 | 1.44 | 0.1655 | Gleasonian |
| All | A | 53 | 233 | 340.4 | 19.3 | 5.56 | <0.0001 | 6,387 | 3,127.8 | 1,032.9 | −3.16 | 0.0016 | 1.66 | 0.0633 | Gleasonian |
| All | W | 63 | 400 | 425.7 | 22.7 | 1.13 | 0.2591 | 6,969 | 4,705.3 | 1,372.5 | −1.65 | 0.0991 | 1.18 | 0.3249 | Random |
| All | F | 56 | 293 | 354.6 | 18.3 | 3.37 | 0.0008 | 5,885 | 2,942.4 | 977.6 | −3.01 | 0.0026 | 1.05 | 0.4264 | Gleasonian |
| Strong | N | 31 | 117 | 149.2 | 9.9 | 3.24 | 0.0012 | 1,428 | 595.7 | 228.9 | −3.64 | 0.0003 | 1.74 | 0.0263 | Clementsian |
| Strong | U | 32 | 127 | 160.4 | 11.2 | 2.99 | 0.0028 | 1,892 | 787.7 | 271.7 | −4.06 | <0.0001 | 2.51 | 0.0003 | Clementsian |
| Strong | E | 31 | 109 | 168.7 | 12.6 | 4.74 | <0.0001 | 3,552 | 1,283.7 | 420.1 | −5.40 | <0.0001 | 2.20 | 0.0019 | Clementsian |
| Strong | T | 21 | 66 | 83.4 | 8.1 | 2.14 | 0.0322 | 192 | 411.9 | 159.7 | 1.38 | 0.1685 | 1.62 | 0.0756 | – |
| Strong | K | 32 | 132 | 167.1 | 12.6 | 2.79 | 0.0053 | 1,712 | 1,303.0 | 392.7 | −1.04 | 0.2976 | 1.83 | 0.0121 | – |
| Strong | A | 24 | 93 | 134.6 | 10.7 | 3.89 | <0.0001 | 1,974 | 1,060.4 | 352.9 | −2.59 | 0.0096 | 0.49 | 0.0894 | Gleasonian |
| Strong | W | 30 | 159 | 184.3 | 14.4 | 1.76 | 0.0784 | 2,341 | 2,052.6 | 592.4 | −0.49 | 0.6263 | 0.52 | 0.0483 | Random |
| Strong | F | 22 | 112 | 117.5 | 9.9 | 0.56 | 0.5755 | 1,036 | 810.8 | 280.1 | −0.80 | 0.4213 | 2.29 | 0.0008 | Random |
| Weak | N | 24 | 119 | 137.8 | 10.8 | 1.75 | 0.0804 | 1,319 | 910.3 | 301.9 | −1.35 | 0.1759 | 0.92 | 0.4207 | Random |
| Weak | U | 33 | 128 | 187.3 | 13.3 | 4.45 | <0.0001 | 3,483 | 1,462.2 | 410.0 | −4.93 | <0.0001 | 0.63 | 0.1168 | Gleasonian |
| Weak | E | 28 | 111 | 156.8 | 11.9 | 3.83 | 0.0001 | 3,469 | 1,239.6 | 385.7 | −5.78 | <0.0001 | 1.96 | 0.0038 | Clementsian |
| Weak | T | 18 | 87 | 89.5 | 8.1 | 0.31 | 0.7599 | 609 | 513.1 | 167.7 | −0.57 | 0.5673 | 1.72 | 0.0107 | Random |
| Weak | K | 31 | 156 | 175.7 | 12.3 | 1.60 | 0.109 | 2,108 | 1,282.5 | 382.4 | −2.16 | 0.0309 | 1.14 | 0.3232 | Random |
| Weak | A | 26 | 113 | 158.0 | 13.3 | 3.39 | 0.0007 | 2,408 | 1,627.6 | 490.9 | −1.59 | 0.1119 | 1.50 | 0.0558 | – |
| Weak | W | 30 | 164 | 190.1 | 13.9 | 1.87 | 0.0611 | 2,775 | 1,840.7 | 560.3 | −1.67 | 0.0954 | 0.74 | 0.2359 | Random |
| Weak | F | 31 | 147 | 192.4 | 13.3 | 3.42 | 0.0006 | 3,861 | 1,806.9 | 557.5 | −3.68 | 0.0002 | 1.18 | 0.2630 | Gleasonian |

Notes:
Results are given for the first axis of reciprocal averaging ordination on the species by site matrices testing for coherence, species range turnover and boundary clumping in each metacommunity of 15 sites across eight regions of New Zealand. Mean and SD values are those calculated from the 1,000 generated null matrices, based on the 'R1' null model. Refer to Fig. 1 for region names. '–' represents structures with non-significant turnover.
Subset, subset of species (all, and strong and weak dispersers); df, degrees of freedom; Abs, number of embedded absences; Re, number of replacements; MI, Morista's Index; SD, standard deviation.

Egmont (Clementsian) and Westland (random) had the same pattern between high and low dispersal ability groups. Tararua consistently exhibited weak patterns with either random or no structure, and Westland metacommunities were always randomly distributed.

## DISCUSSION

As a result of the relatively high latitude of New Zealand and based on the hypotheses of *Jocque et al. (2010)*, we hypothesised ($H_1$) a dominant role of species sorting and dispersal surplus (reflecting the mass effects archetype) in structuring these assemblages ($H_{1a}$) and an increasing dispersal surplus from north to south ($H_{1b}$). However, despite a latitudinal gradient present in assemblages at the community level overall and within each

island for regional $\gamma$ diversity (as well as $\alpha$ and $\beta$ diversity, *Astorga et al., 2014*), what emerged at the metacommunity level was more idiosyncratic. In particular, there was no latitudinal trend in either environmental vs. spatial control (rejecting $H_{1b}$) or the idealised metacommunity types tested through the EMS analysis at both the full community level and for dispersal groups. Lack of fit to the hypothesis of *Jocque et al. (2010)* likely reflects the dynamic, unpredictable nature of New Zealand streams (partially supporting $H_1A$).

New Zealand comprises a series of mid-latitude islands, with a typically unpredictable climate (Fig. 1) and flashy river flow regimes (*Winterbourn, Rounick & Cowie, 1981*) reflecting its oceanic position. At a single time-point, communities are therefore most likely at different stages of post-flood recolonisation ($H_1A$). Antecedent conditions are not only important for dynamic systems like these, but also for more continental climates. For instance, preceding-year climatic conditions have been demonstrated to be more important in shaping European stream invertebrate communities than long-term climatic trends (*Jourdan et al., 2018*). The dynamism of streams, particularly in oceanic climates, represents a fundamentally important factor controlling metacommunity dynamics, with assembly mechanisms varying temporally in dynamic streams (*Datry, Bonada & Heino, 2016*; *Sarremejane et al., 2017*). The relative roles of local and regional processes will depend on the amount of time that has passed for dispersal and colonisation to play out (*Brendonck et al., 2014*). With the central importance of natural cycles of flooding and drought in streams (*Poff et al., 1997*; *McMullen et al., 2017*; *Tonkin et al., 2018b*), it stands to reason that antecedent flow conditions play a key role in structuring metacommunities in streams (*Campbell et al., 2015*).

The lack of seasonality and predictability in New Zealand's climate likely plays a strong role in the low predictability in metacommunity structuring. The hypothesis of *Jocque et al. (2010)* does not take into account differences in island size and isolation, fundamental aspects controlling biodiversity (*MacArthur & Wilson, 1967*). Island and mainland locations at similar latitudes do not comprise the same climatic patterns (*Tonkin et al., 2017a*), with continental locations having much greater predictability in their seasonality compared to islands. To demonstrate this point, we compared a 30-year sequence of monthly rainfall totals from the central North Island of New Zealand with Western Australia, a Mediterranean climate, using wavelet analysis (Fig. 1) (*Torrence & Compo, 1998*). Although this is just one of the locations examined in our study, which vary in their rainfall regimes, we use this simple comparison to demonstrate the extent of climatic unpredictability present in this region compared to a predictable climatic zone. Figure 1 demonstrates clearly the strongly seasonal and predictable pattern apparent in Western Australia, with a significant and repeatable cycle at the one-year time period over the full sequence. By contrast, central New Zealand's climate exhibits no repeatability in the rainfall, with very few time points in the sequence indicating any power at the one-year period.

New Zealand streams have other features that may limit their fit to our primary hypotheses, some of which are shared by other island localities, including: rivers tend to be short, swift, and steep due to the narrow landmass and tectonically active nature;

evergreen vegetation dominates the flora; and riparian vegetation is scarce for much of their length leading to a predominance of autochthonous rather than allochthonous control of river food webs (*Winterbourn, Rounick & Cowie, 1981*; *Thompson & Townsend, 2000*). As such, New Zealand streams are considered as being physically, rather than biologically, dominated systems (*Winterbourn, Rounick & Cowie, 1981*). These factors, in conjunction with its highly dynamic geological history, have led to the evolution of a stream invertebrate fauna with flexible and poorly synchronised life histories, and generalist feeding behaviour (*Winterbourn, Rounick & Cowie, 1981*; *Thompson & Townsend, 2000*; *Scarsbrook, 2000*). Although New Zealand stream invertebrate communities are not necessarily less species rich or different in terms of food web structure to overseas locations, there is a clear paucity of shredder species in particular, with generalist browsers predominating communities (*Thompson & Townsend, 2000*). Under these circumstances, it is not surprising that metacommunity dynamics can be difficult to predict, as we clearly demonstrate, without a strong temporal resolution in the data. Thus, in support of our alternative first hypothesis, despite the large latitudinal gradient examined, predictable metacommunity dynamics appear to be masked by short-term unpredictability in environmental conditions.

Results were highly idiosyncratic between different regions, with considerable variability in the relative roles of environmental and spatial structuring, important environmental variables, and the idealised metacommunity types, with no real match between the two approaches. This context dependence did not reflect an interaction between spatial extent and dispersal ability. Although much of this unpredictability may be related to the unpredictable characteristics of New Zealand streams, it is pertinent to recognise that this is a challenge facing many stream metacommunity studies globally, where patterns differ considerably between different catchments (*Heino et al., 2012*, *2015a*; *Tonkin et al., 2016a*). *Lawton (1999)* pinpointed this problem in ecology over a decade ago suggesting that community ecology is rife with contingency, so much so that generality is unlikely. Lawton goes on to highlight that the problem is indeed most severe at the intermediate organisational level of communities, compared to more predictable lower (e.g. populations) or higher levels (e.g. macroecology). Metacommunities are indeed difficult systems to predict, with processes affecting different subsets of organisms and operating at specific times (*Driscoll & Lindenmayer, 2009*). One source of context dependence in metacommunity structuring is differences between different trait modalities, such as dispersal modes (*Thompson & Townsend, 2006*; *Canedo-Arguelles et al., 2015*; *Tonkin et al., 2016b*). Thus, if spatial extent and dispersal limitation were interacting to structure the metacommunity, deconstructing the full assemblage into dispersal groups (e.g. strong vs. weak dispersers) should have helped to explain discrepancies in our predictions, but this was not the case. Nevertheless, we must also entertain the possibility that greater spatial replication would have strengthened the observed patterns.

Finally, contrary to the expectation of *Jocque et al. (2010)* that dispersal ability increases moving away from the equator ($H_2$), we found a decrease in the ratio of strong to weak dispersers moving from north to south. Theoretically, temporal variability in environmental conditions promotes increased dispersal ability of organisms

(*Dynesius & Jansson, 2000*; *Jocque et al., 2010*); an hypothesis strongly tied with Rapoport's rule of increasing range size with increasing latitude (*Stevens, 1989*) and one that receives support from the population genetics literature via increased genetic divergence among populations nearer the equator (*Eo, Wares & Carroll, 2008*). However, it is important to note that while dispersal ability can play a strong role in determining species range sizes, its influence may be less common than previously thought (*Lester et al., 2007*). Although there is evidence that weak dispersers have stronger latitudinal diversity gradients than strong dispersers in Europe, the mechanisms behind this are related to the ability of organisms to recolonise northern sites following glaciation (*Baselga et al., 2012*); a different issue to that experienced in New Zealand. The conflicting result we observed may reflect several factors. (1) Lack of time for dispersal and colonisation to play out post-disturbance (*Brendonck et al., 2014*; *Campbell et al., 2015*). (2) The requirement of a longer latitudinal gradient for these mechanisms to play out. Over the length of New Zealand, the continuity of habitat availability in space and time, a key mechanism behind *Jocque et al. (2010)*, likely differs very little. (3) Climatic idiosyncrasies not reflecting a north–south gradient and thus not selecting for a gradually increased dispersal ability at higher latitudes.

## CONCLUSIONS

*Jocque et al. (2010)* highlighted the fundamental role of dispersal in driving the latitudinal diversity gradient, suggesting a climate-mediated dispersal–ecological specialisation trade-off as a key factor regulating this pattern. We tested several hypotheses based on those of *Jocque et al. (2010)* relating to how New Zealand stream invertebrate metacommunity structure changed along a broad latitudinal gradient, and examining the mediating role of dispersal. We rejected our primary hypotheses, finding that: (1) species sorting appears to be weak or inconsistent, and its influence did not change predictably with latitude; and (2) weaker dispersers increased with latitude. We associate this lack of fit to these hypotheses on the strong unpredictability of New Zealand's dynamic stream ecosystems (supporting $H_1A$) and a fauna that has evolved to cope with these conditions. While local community structure turned over along this latitudinal gradient, metacommunity structure was highly context dependent and dispersal traits did not elucidate patterns.

These results, along with other recent findings (*Heino et al., 2012*, *2015a*; *Tonkin et al., 2016a*), provide a cautionary tale for examining singular metacommunities. The considerable level of unexplained context dependency suggests that any inferences drawn from one-off snapshot sampling may be misleading. Given the importance of understanding metacommunity processes for the successful management of river ecosystems (*Siqueira et al., 2012*; *Heino, 2013*; *Tonkin et al., 2014*; *Stoll et al., 2016*; *Swan & Brown, 2017*), this level of unpredictability is a major cause for concern. While spatial replication of multiple metacommunities may elucidate some of this uncertainty, studies on temporal dynamics of metacommunity processes are clearly needed. We therefore urge researchers to consider the temporal dynamic, particularly in relation to seasonal cycles and their predictability.

## ACKNOWLEDGEMENTS

We thank Fiona Death, Manas Chakraborty and Riku Paavola for field and laboratory assistance. Jenny Jyrkänkallio-Mikkola, Félix Picazo, and two anonymous reviewers improved earlier versions of the manuscript.

### Funding

The authors received no funding for this work.

### Competing Interests

Jonathan D. Tonkin is an Academic Editor for PeerJ.

### Author Contributions

- Jonathan D. Tonkin conceived and designed the experiments, analyzed the data, prepared figures and/or tables, authored or reviewed drafts of the paper, approved the final draft.
- Russell G. Death conceived and designed the experiments, authored or reviewed drafts of the paper, approved the final draft.
- Timo Muotka performed the experiments, authored or reviewed drafts of the paper, approved the final draft.
- Anna Astorga performed the experiments, authored or reviewed drafts of the paper, approved the final draft.
- David A. Lytle conceived and designed the experiments, authored or reviewed drafts of the paper, approved the final draft.

### Data Availability

Tonkin, Jonathan D; Death, Russell G; Muotka, Timo; Astorga, Anna (2018): Stream invertebrate data and local habitat variables from 120 New Zealand streams. figshare. Fileset. https://doi.org/10.6084/m9.figshare.5917267.v1.

### Supplemental Information

Supplemental information for this article can be found online at http://dx.doi.org/10.7717/peerj.4898#supplemental-information.

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
