# Peer review of "Do latitudinal gradients exist in New Zealand stream invertebrate metacommunities?"

_PeerJ, doi:10.7717/peerj.4898_

## Round 0.1 · original submission · Minor Revisions

Here I received comments from two reviewers. They both were highly positive and suggested minor revision. Overall, I found most of their comments could be followed. I have no additional comments and expect to see the authors' revised version soon.

·

Basic reporting

The manuscript is clear and precise and written with close to flawless English. The structure follows the journal’s standards as well a good academic writing and findings are stated in concise manner. In general, the figures and tables are relevant, clear and well structured and raw material is provided. The scope of the study is well stated in the introduction and relevant references are cited.

Experimental design

The study is applicable to the scope of the journal. Both the biological and environmental data sets are large enough to study the topic. The design supports the research questions. The methods are described thoroughly .The chosen statistical methods are applicable to the research questions and the results are reported in a meaningful way.

Validity of the findings

The authors address a highly important topic of the factors affecting freshwater metacommunity structure. There is an ongoing need to understand how communities of dynamic freshwater environments are structured in order to conserve freshwater organisms. The authors discuss the importance of temporal factors affecting metacommunities in fluvial systems, which I consider to be one of the key factors to take into account in studies focusing on these environments. The authors also discuss about the uniqueness of catchments within region, which is another important thing to acknowledge in my view.

Additional comments

Specific comments

Introduction

page 3, line: ‘island size’ comes up somewhat out of nowhere here. Maybe change the order of this and next sentence.
page 3, line 45: please change ‘drives’ with ‘drive’
page 3, line 54: I was very interested to read what Jocque et al think are the reasons for this. Maybe add a sentence of this.
page 3, line 58: How does this unpredictability affect the examination of latitudinal factors on communities? Authors do address this quite well in the discussion but maybe some of it should also be discussed here.
page 4, lines 72-74: Could the authors explain here how latitudinal diversity gradient is linked to dispersal ability.
page 4, line 75: word ‘this’ seems redundant

Material and methods

page 5, lines 101-102: Why were sites with higher human impact not included and will the results found in the study apply to such environments? Including such environments might have broadened the range of the measured environmental variables, which might have resulted in stronger species sorting. What is the authors’ view on this?
page 7, line 158: This could be included as last sentence instead of placing it here as a separate sentence.
page 5, lines 109-110: Please consider changing the structure of the sentence. Maybe: “Benthic macroinvertebrate sampling was performed between February and April 2006 using...”
page 5, line 168: Please consider changing the word ‘completeness’ to something else. Surely one cannot sample everything from a site.
page 8, line 193: Couldn’t there be spatially structured environmental variables also in the pure spatial effects? Were the environmental variables spatially autocorrelated and was this tested?
page 8, line 202: Were the sampling sites within a region connected with same river network?
page 9, line 215: I am surprised that the stream order did not correlate with any of the measured environmental variables. Was this really the case?
page 9, lines 229-230: This paragraph contains only one sentence. Please consider joining with some other paragraph
page 11, line 284: Please change ‘neseted’ to ‘nested’.
page 12, lines 289-290: Yes, but also on the range of the measured environmental variables.

Results

page 12, 306-307: I like the figure as it presents a clear difference in the communities. However, through the figure one cannot clearly state that this is because of latitude and not some other factor. Please consider revising the sentence.
page 12, lines 308-309: Is the reason for this discussed?

Discussion

pages 14-15, lines 366-368: Can this be understood so that studying the effect of climatic factors on communities in such environments may not be reasonable? Does this finding reflect the concept of each stream harboring unique conditions which may question the categorization of streams based purely on latitude?

page 16, lines 412-413: If samples per region would have been larger, would similar results have been observed? What is the authors’ view on this?
Table 1: Maybe the table would be better located in the supplementary material.

·

Basic reporting

This paper focuses on metacommunity processes as driver of latitudinal diversity patterns. I find this study a relevant contribution to debate on latitudinal gradients of diversity as most previous literature approach it from community level. In addition, most of those studies did it in the north hemisphere, the southern hemisphere remaining poor studied. This is a point that authors should remark in the introduction section in order to increase the interest of the manuscript. It is overall well written and the English writing is correct. However, I suggest them to take a last quite reading to solve some minor mistakes. I consider the background is somewhat incomplete and authors should mainly improve it on latitudinal patterns of diversity (and provide new references). The part concerning metacommunity issues is good enough but given it is framed on latitudinal patterns authors must improve the background for this part. The structure of the article matches the standard sections and figures are of high quality, relevant to the content of the article and appropriately labelled. I find the addressed results are relevant to the hypotheses stated. However, I have some concerns which are detailed in the "General comments for the author".

Experimental design

Research questions are well defined and are relevant enough to get publication. I think authors should better highlight the knowledge gap they are investigating as it is explicitly stated in first sentence of the abstract. I find this piece of research to be rigorously conducted with high technical standard. Methods are well described and information on them is good enough. In fact, I find this section too extensive (introduction + results + discussion take 7 pages while methods by itself takes 7,5 pages), and this the reading of the paper a little tedious. In order to solve this point, given that EMS analysis is used as an additional exploratory analysis and their results do not provide clear patterns and relevant results, I suggest authors to briefly mention this point in terms of methods, results and discussion and extend on it in one appendix. In this way, they can significantly reduce the methods section, as this point takes 2,5 pages of it.

Validity of the findings

In my opinion, the impact and novelty of the paper should be more clearly stated. Although I consider data are robust enough and statistically sound, I encourage authors to share data and R code. At least, a table with taxa list and another with local, metacommunity and regional richness should be provided. Conclusions are well stated and adequately linked to the research questions.

Additional comments

Abstract

I find three strong statements in the abstract which should be better supported along the manuscript: i) unpredictability of New Zealand’s dynamic stream ecosystems (please, provide data on environmental features of the study area); ii) fauna that has evolved to cope with these conditions (please, provide information on proportion of endemics-biogeographic traits-range size); iii) while local community structure turned over along this latitudinal gradient, metacommunity structure was highly context dependent and dispersal traits did not elucidate patterns (please, show in a clear way the inverse latitudinal pattern at community level).

Introduction

Please, provide references and background about latitudinal patterns and nuances for other groups/regions. Overall, I miss a paragraph briefly reviewing the extensive literature on latitudinal patterns, and specifying that although a general pattern can be found, there are many exceptions mainly depending on the origin and adaptations of the targeted group of organisms (see general reviews by Hillebrand 2004 in AmNat; Heino 2011 in Freshwater Biol; book “Patterns and processes in macroecology” by Gaston & Balckburn 2000; as well as concrete studies as for example Dobrovolski et al 2011 in GlobalEcolBiogeogr; Hortal et al 2011 in EcolLett; Hof et al 2008 in GlobalEcolBiogeogr; Boyero et al 2012 in GlobalEcolBiogeogr; etc.). Authors should also mention the role of random factors driving latitudinal patterns as the main domain effect (Colwell & Lees 2000 in TrendsEcolEvol). Are there studies for latitudinal patterns in New Zealand? If yes, please provide some references. If not, it is worth you show the latitudinal pattern for regional and local richness with your data. It is interesting for comparson with the metacommunity latitudinal pattern. Authors should also mention that most studies on latitudinal patterns have been performed in North America and Europe, their main findings pointing to dispersal ability in combination with climatic legacies from Pleistocene glaciations as important drivers of latitudinal patterns (again, please mention some references as those I provided above).

Line 40: Please, provide some reference to support you statement about the influence of seasonality and predictability on latitude-biodiversity relationship.

Line 40-41: I feel authors should first briefly lists the main described mechanisms driving the latitudinal patterns. In the last two decades there has been an intense debate about drivers of latitude-biodiversity relationships and how they interact, and multiple hypothesis have been put forward as possible explanations. Most prevalent among them are those related to energy-water availabitiliy, area size, biological interactions, past environments and effective evolutionary time (e.g. Ricklefs 2004 in EcolLett; Pianka 1966 in AmNat; Willig et al. 2003 in AnnuRevEcolEvolSyst; Mittelbach et al 2007 in EcolLett).

Line 43: I feel “similarly placed” is a little ambiguous. Does it mean very close in terms of geographical distance, same latitudinal range, both? Please, concrete.

Lines 62-65: Although you provide some references, I think it is a key point to understand the context of the study hypotheses and findings, so I recommend to extend a little on geology, biogeography and climatic history particularities of New Zealand as well as on the general features of its stream fauna. What main groups of organisms form stream benthic fauna in New Zealand? Provide some basic data about its origin. Are they mainly widespread species arriving from the continent? Have they evolved to new local species resulting in a high proportion of endemics adapted to the special features of New Zealand streams? Very little is known about latitudinal patterns in New Zealand, so I think some context about these points would help to increase the understanding on the relevance of this study by making it more comparable with the extended literature dealing with latitudinal patterns in North America and Europe.

Methods

Lines 99-100: “Site selection followed a series of criteria to minimise differences between regions”. Do authors refer to the points stated in the following sentence? I guess so, but for me it is not clear enough as it is currently written.

Data for analyses: although authors mentioned it for the nMDS analysis, I think they should state more clearly if, overall, all analyses are based on presence-absence or abundance data. Please, specify.

Line 126: “previously-identified important local habitat variables”. How was it done? Is it taken from previous references or authors performed some analysis? Please, specify.

Line 220: It should be “by Blanchet et al., (2008)” instead of “by (Blanchet et al., 2008).

Line 221-223: I find this sentence a little confusing. Please, rewrite.

Line 247: Please, specify the number of permutations you used.

Lines 248-249: I find this sentence a little confusing. Please, rewrite.

Lines 298-300: Good point. I find this approach very interesting. In terms of community ecology, I am not sure if it is correct to remove species based only in geographic rarity, i.e. rarity of occupancy (as authors did), or should be this species removed only when, in addition to geographic rarity, they also show demographic rarity (rarity of individuals within areas or density rarity). I mean, if one species is only found in one place but display high abundance or even adults and larvae are found in the same place, this species can be totally considered as a member of the community. What about community structure? Does it depend on the total number of sites forming the metacommunity? I think this sentence could be extended a little on this point.

Results

Line 302: Are data on spatial extents of metacommunities given in any place of the manuscript?

Line 305: I think authors should provide information about the variables that mainly summarise these first to axes.

Line 307: I suggest to complement results of PERMANOVA with a table/plot showing the latitudinal trend of local and regional richness. Showing the latitudinal pattern of local and regional richness will also help to interpret the results at metacommunity level. As authors have community data and there is a biogeographic regionalization of New Zealand, it would be very easy to perform it and I am sure it will provide a very clear and visual insights of latitudinal diversity patterns of the study area. I consider this point relevant to researchers working on latitudinal patterns and could help your paper be mentioned in further research on this topic.

Line 308: Does author mean that the latitudinal pattern for the whole study area is broken and divide into two similar patterns for each single island? I think this is an interesting point on which author should go in depth in the discussion section.

Lines 309-311: Authors should provide data on percentage of completeness for each accumulation curve. How they support this statement? How could readers know if the regional pool is well sampled or not? They should also provide some reference on what could be considered adequate inventories for invertebrates (see for instance Sánchez-Fernández et al. 2008 in DiversDistributions).

Lines 311-312: I think this is an interesting point that could be related to the strong-weak dispersal ratio. I suggest authors to write on it in the discussion section. For strong dispersers, regional curves are expected to reach the asymptote faster than for weak dispersers. As it matches the results, authors could relate dispersal with range size and biogeography of species and therefore with metacommunity structure and dynamics. It could also encourage authors to introduce some sentence dealing with the relevance of their findings in terms of conservation as this is a point that I missed in the paper and could increase its relevance.

Lines 318-319: Very relevant finding. Authors should definitely discuss on it as most of the studies on latitudinal patterns in Europe and North America find poor dispersers to be accumulated in lower latitudes whereas strong dispersers predominate in high latitude areas. Please, provide also references.

Discussion

Lines 375-377. In order to increase the interest of the paper for a general audience, authors should provide some relevant information (maybe in the introduction section) on the features of the stream studied, mainly in terms of temporal dynamics (flow regimes, temporality or intermittency, influence of droughts and floods, etc). In the current state of the manuscript, it is only stated in a very sligh way.

Lines 405-406: As previous studies have found nuances in the latitudinal patterns depending on the targeted taxonomic group, authors should write a little about particularities of stream invertebrate fauna from New Zealand (at least at order level, they work on the whole community). This can provide very useful and relevant information for researchers working on this topic what could be mentioned in next studies.

Lines 442-443. Authors should provide information on climate patterns of New Zealand in some place along the manuscript (maybe in the methods section), because when a reader arrives to this point it is not clear yet if there is a clear climate gradient and how much can it influence the nature of New Zealand streams (temporality, intermittence, etc.). This is a relevant point to understand the idiosyncrasies of benthic stream faunas. Maybe a map on this point would be great (if not, at least some reference).

Conclusion

Lines 453-454. In order to better understand this point, as I mentioned above, it should be very interesting that authors provide some key aspects about biogeographic traits of species (at least, how is the proportion of endemics vs widespread species?).

Figures

Figure 2. I suggest to include the boundaries of the five biogeographic regions in the map and also change the symbols of sampling sites for each metacommunity (manly to get a better readability of PCA and nMDS plots). Maybe it is better to insert the letters for each metacommunity in the map. Please, provide % of completeness for accumulation curves. I guess letters for each metacommunity in several plots follow a latitudinal order from low to high latitude, but maybe it should be specified in any place (alternatively they can be ordered according to their latitudinal centroids).

---

## Round 0.2 · accepted · Accept

Overall, I agree with the reviewers and the authors did a good job incorporating reviewer comments and addressing the key issues with the manuscript. It is an excellent contribution to the field.

# ·

Basic reporting

The authors have answered my questions adequately and made all relevant corrections. I have no more comments for this manuscript.

Experimental design

No additional comments needed

Validity of the findings

No additional comments needed

Additional comments

I congratulate the authors for a very interesting paper.

·

Basic reporting

I have now had the opportunity of reviewing the new version of the manuscript. Overall, I feel happy about the effort made by authors to modify the manuscript based on the suggestions made by the other reviewer and myself and to argue when these suggestions were not taking into account. Please, if they want to include my name in the Acknowledgements section, substitute "Felix Picazo Mota" by "Félix Picazo".

Experimental design

I have no additional suggestions on this point.

Validity of the findings

I have no additional suggestions on this point.